# Intercalated water layers promote thermal dissipation at bio–nano interfaces

Yanlei Wang[1], Zhao Qin[2,3], Markus J. Buehler[2,3] & Zhiping Xu[1]

The increasing interest in developing nanodevices for biophysical and biomedical applications results in concerns about thermal management at interfaces between tissues and electronic devices. However, there is neither sufficient knowledge nor suitable tools for the characterization of thermal properties at interfaces between materials of contrasting mechanics, which are essential for design with reliability. Here we use computational simulations to quantify thermal transfer across the cell membrane–graphene interface. We find that the intercalated water displays a layered order below a critical value of ∼1 nm nanoconfinement, mediating the interfacial thermal coupling, and efficiently enhancing the thermal dissipation. We thereafter develop an analytical model to evaluate the critical value for power generation in graphene before significant heat is accumulated to disturb living tissues. These findings may provide a basis for the rational design of wearable and implantable nanodevices in biosensing and thermotherapic treatments where thermal dissipation and transport processes are crucial.

[1] Applied Mechanics Laboratory, Department of Engineering Mechanics and Center for Nano and Micro Mechanics, Tsinghua University, Beijing 100084, China. [2] Laboratory for Atomistic and Molecular Mechanics, Department of Civil and Environmental Engineering, Massachusetts Institute of Technology, 77 Massachusetts Avenue, Room 1–290, Cambridge, Massachusetts 02139, USA. [3] Center for Computational Engineering, Massachusetts Institute of Technology, 77 Massachusetts Avenue, Room 1–290, Cambridge, Massachusetts 02139, USA. Correspondence and requests for materials should be addressed to Z.X. (email: xuzp@tsinghua.edu.cn) or to Z.Q. (email: qinzhao@MIT.EDU) or to M.J.B. (email: mbuehler@MIT.EDU).

Recently, there have been increasing efforts reported to develop bio-nano devices for biophysical and biomedical applications[1–6]. Functional nanodevices are able to detect the biophysiochemical signals, gather valuable information from reactions of biological significance and manipulate cellular activities through the interface between biological tissues and functional nanostructures made of synthetic materials[1–6]. The operation of these devices is usually accompanied with significant heat generation that is localized at an extremely small length scale[7]. Moreover, the devices made of synthetic nanostructured have material properties that are significantly different from biological tissues, which could thus introduce a thermal barrier accounting for easy heat accumulation to disturb the living system. The limited knowledge about thermal management at the interface between biological tissues and synthetic nanostructures has resulted in a lot of concerns about the device performance and the interfacial thermal coupling is no doubt pivotal in designing relevant applications. On the other hand, intentional thermal management of biological tissues has also been demonstrated as an effective way in controlling gene express, tumour metabolism and cell-directed treatment of diseases[8–11]. Key questions that have not been addressed for these applications include how the energy inputs into devices induce local temperature rise at such a nanoscale interface and what is the criterion of thermal stability for the living systems not to be perturbed, or for the thermoregulation to be activated[12]. However, neither enough knowledge nor a model is available at present to evaluate the interfacial thermal coupling, to answer the abovementioned questions, and it is difficult for experiments to directly characterize this interfacial thermal transport and dissipation processes, as it requires both high spatial and temporal resolution and measurement in a solvent environment.

Graphene and its derivative graphene oxide (GO) are flexible films of single-atom level thickness that are able to form a close contact with lipid layers to cover the cell membrane[2–4,13–15]. Although recent studies show that micro- and nanometre-sized graphene sheets could enter the hydrophobic interior of biological membranes[16,17], our work here focuses on the application of graphene–membrane contacts for sensing and actuation applications, where the graphene sheet covers the cell surface. As an example of this type of bio–nano interface, nanocomposite structures have been constructed by depositing lipids onto graphitic layers, or intercalating them into an assembly in a layer-by-layer manner[2–4,13–15]. In earlier work, researchers have found that graphene layers could confine cells as an easy-to-apply impermeable and electron-transparent encasement that retains the cellular water content[4]. The

graphene sheet deposited onto the plant cell wall could also permit a free functioning of the plasma membrane it encapsulates[3]. The stability of these interfacial structures suggests that graphene can be considered as a promising material for nanodevices due to its specific interactions with biological tissues. However, the Young's modulus of graphene is approximately six orders of magnitude higher than that of a cell membrane and their vibrational modes are drastically different, which consequently lead to a high thermal barrier for local heat accumulation to build up and breaks down the device performance. Moreover, evidences have been reported for the existence of a layer of trapped water between graphene and lipid bilayers, which could mediate the dielectric coupling across the interface and thus the disruptive effects to the cellular membrane[2,13]. In addition to this electromechanical coupling effect, interfacial thermal energy transfer through the graphene/water/lipid contact is also vital for both reliable functioning of nanodevices and stability of the living system. This fact, however, has not been explored yet to the best of our knowledge.

In this work, we use large-scale fully atomistic molecular dynamics (MD) simulations to investigate thermal conduction at the interface between lipid bilayers and graphene or GO sheets. We focus on how the intercalated water layer affects the thermal coupling at the interface and how it further determines the critical power density of heat generation in graphene, above which the temperature in lipid membrane will be elevated beyond that in the environment. We find that the intercalated water layers play a critical role in modulating the interfacial thermal coupling by decreasing the energy that flux into lipid bilayers, and promoting the interfacial thermal conductance (ITC). The ITC for this 'soft' interface is weaker compared with that of the 'hard' interfaces such as graphene–silica and graphene–metal, but is at the same level as many other solid–polymer and solid–liquid interfaces. On the basis of these findings, we develop a predictable model to estimate the temperature evolution in graphene and the lipid bilayer under a certain power density, respectively, and discuss the implication of our results in the design of nanodevices for relevant applications.

## Results

**Molecular structures and water diffusivity.** As depicted in Fig. 1a, our model is composed of a lipid bilayer and a monolayer graphene or GO, which are separated by a certain distance of ~1 nm. The composite structure is immersed in water. As reported in the previous studies, there is a thin layer of trapped water at its interface, which can be finely tuned by the humidity

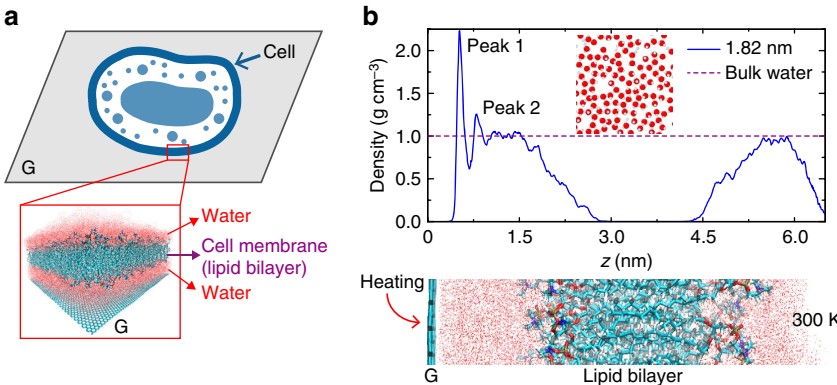

**Figure 1 | Graphene–water–lipid bilayer hybrid as a model system.** (**a**) Illustration and a simulation snapshot of the model under exploration. (**b**) The mass density profile of water molecules plotted with that of bulk water, along the distance measured from graphene (top of panel). The simulation set-up to explore heat transfer across the interface (bottom of panel), which is aligned with the density profile plot. The thickness of intercalated water layer is $t_W = 1.82$ nm.

of experimental conditions[18,19]. Thus, we develop our computational model by filling the interfacial gap with water molecules. The molecular structures are equilibrated in our MD simulations at ambient condition. From the spatial density profiles of water molecules, we identify structured water layers adjacent to the graphene wall. For example, for the intercalated water layer with a thickness of $t_W = 1.82$ nm, the profile showed in Fig. 1b features two prominent peaks (peak 1 and 2), which demonstrate the layered structure. Here $t_W$ is defined as the span of region where the mass density of water is equal or higher than that in the bulk phase. The amplitudes of peak densities are higher than that of bulk water, which is a common feature of nanoconfined water and was reported for water trapped between graphene or GO layers at a similar length scale of spatial confinement[20]. From Fig. 1b, we also find that there are water molecules entering the lipid bilayer at a depth of $\sim 1$ nm, with a smoothly decaying amplitude of density. From an application viewpoint, the presence of intercalated water layers not only modulates the mass and energy delivery between bio- and nanostructures but also could perturb the biological activity of cellular membranes due to its difference with the regular extracellular environment.

Recent experimental measurements suggest that at the graphene–lipid bilayer interface, the thickness of intercalated water layer between the lipid membrane and graphene is on the order of a few nanometres[2]. We thus explore a number of interfaces with $t_W$ ranging from 0 to 2 nm. Figure 2a shows that there is a notable difference between the density profiles of water at graphene and GO interfaces in both the amplitudes and positions of peaks, which is caused by the presence of charged functional groups in GO. From the correlation between $t_W$ and the mass of water under the first density peak, $M_1$ (Fig. 2b), we

conclude that for both graphene and GO, water in the central region of interfacial gallery has a similar structure as bulk water with thickness $t_W$ above $t_{Wc} = 1$ nm, and the magnitude of the first peak becomes independent on the amount of trapped water.

The presence of water in the nanogap between lipid bilayer and nanostructures allows the cell to exchange masses and energy with their surroundings through molecular diffusion, and thus are crucial for their survival. We further show that the molecular diffusivity of interfacial water is significantly lower than that in the bulk (Fig. 2c,d). The presence of bulk-like water at the bio-nano interface instead of structured water[20] is important to maintain the biological activity of cell. Consequently, the value of $t_W$ should be assured to be larger than the critical value $t_{Wc} = 1$ nm, which could be controlled by, for example, tuning the humidity. On the other hand, one should also keep in mind that, to maintain active communication between lipid bilayer and nanostructures for effective electrical signal and energy exchange, the thickness of intercalated water layer has to be kept also below a critical length scale of $\sim 1$ nm[2,3].

**Thermal dissipation at the bio–nano interface.** We then explore the heat dissipation process based on the simulation set-up shown in Fig. 1b. We heat up the graphene or GO layer at a constant power density $p$. As shown in Fig. 3a, at a low heating power density, for example, $p \sim 1$ GW m$^{-2}$, the heat generated can be efficiently dissipated across the interface, along the thermal gradient. No significant destruction of the bilayer structure at this rate of thermalization is observed in our MD simulations that reach steady states. Accordingly, temperature rise in the lipid bilayer is negligible and is almost power-density-independent.

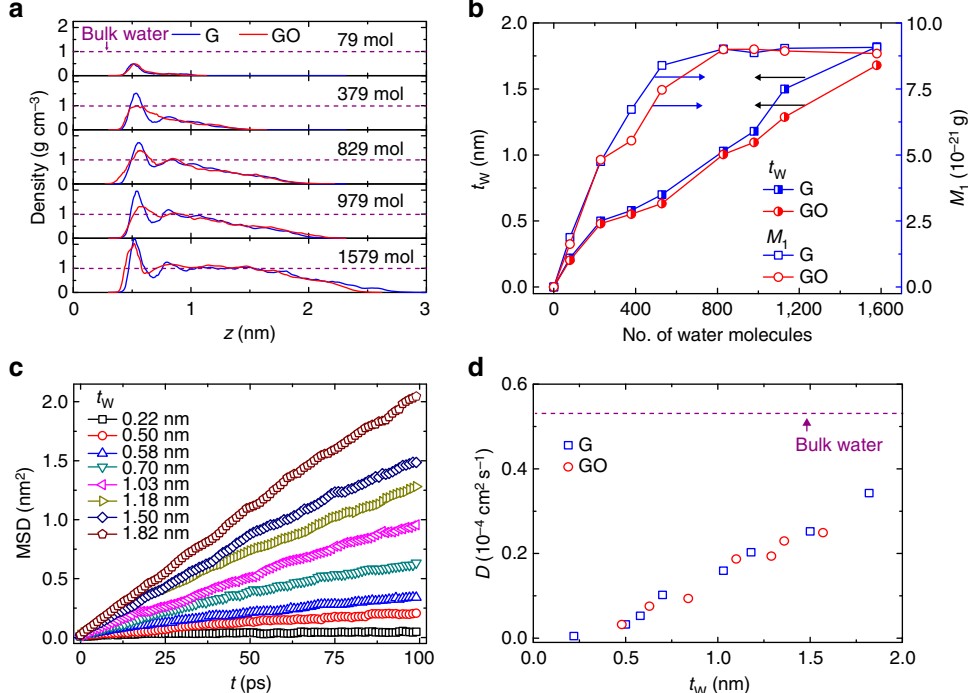

**Figure 2 | Structural and dynamical properties of the intercalated water layers.** (**a**) Mass density profiles of water molecules along the direction perpendicular to the interface, plotted for different numbers of intercalated water molecules. (**b**) The thickness of intercalated water layer $t_W$ and the water mass under the first peak ($M_1$) in the density profiles, plotted as a function of the number of intercalated water molecules. The latter one represents the number of water molecules in the first nearest neighbours. $M_1$ is calculated as $M_1 = \int_{z_0}^{z_1} A\rho(z)\mathrm{d}z$, where $\rho(z)$ is the mass density of water molecules, $A$ is the area of interface, and $z_0$ and $z_1$ are the boundaries of the first peak in the density profiles. (**c,d**) The mean-square distance (MSD) and diffusion constant $D$ measured for the intercalated water layer. $D$ is compared with the diffusivity of bulk water calculated using the same water model[58].

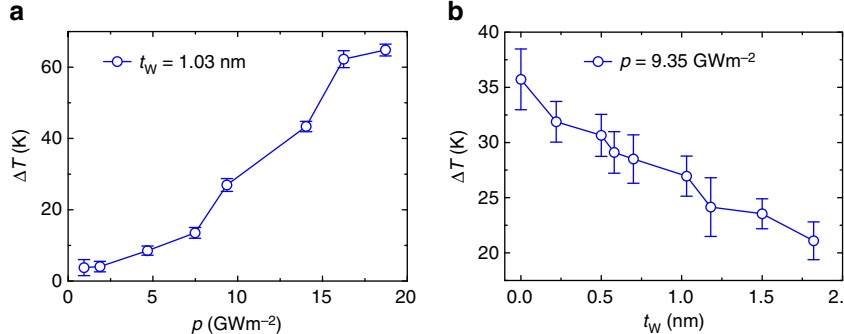

**Figure 3 | Temperature rise in lipid bilayers on heat generation in graphene.** (**a**) The rise in temperature, $\Delta T$, in the lipid bilayer after heat flux with power density $p$ is injected into graphene. A critical power density $p_c$ of $\sim 7.5\,\mathrm{GW\,m^{-2}}$ can be identified for a targeted threshold of temperature rise, $\Delta T_c$. Here we consider $\Delta T_c = 20\,\mathrm{K}$ as an example for illustration. (**b**) Temperature rise in the lipid bilayer as graphene is heated at a power density of $p = \sim 9.35\,\mathrm{GW\,m^{-2}}$, plotted as a function of $t_W$. Error bars are plotted based on results from five independent simulation runs.

However, as the power density exceeds a critical value of $p_c$, the temperature rise in lipid bilayer, $\Delta T$, increases drastically and features an almost linear dependence on the power density (Fig. 3a; Supplementary Fig. 1a). The modest deviation from linearity may arise from the change in the structure and diffusivity of intercalated water at different heating power density, which is more significant for GO because of its stronger interaction with water. Specifically, at $p = \sim 16.25\,\mathrm{GW\,m^{-2}}$, the temperature rise in lipid bilayer could reach $\sim 43\,\mathrm{K}$ within 400 ps before it reaches a plateau (Supplementary Fig. 2) that is significant enough to perturb the physiological behaviour of biological systems, while the temperature rise in graphene is $\sim 1,150\,\mathrm{K}$ that could break down the electronic device performance[21].

One can then define the value of $p_c$ for a targeted threshold value, $\Delta T_c$. The calculated value of $p_c$ thus increases with $t_W$. That is to say, the presence of intercalated water layer significantly enhances heat dissipation and maintains thermal stability of the bio–nano interface (Fig. 3b; Supplementary Fig. 1b). Without loss of the generality, we choose $\Delta T_c = 20\,\mathrm{K}$ according to the reported value of critical temperature rise for a cell to maintain its viability[22,23]. For the intercalated water layer with a thickness of $\sim 1\,\mathrm{nm}$, which is an ideal value considering the competition between molecular diffusivity and cross-interface communication as we discussed earlier, we measure $p_c = \sim 7.5\,\mathrm{GW\,m^{-2}}$ for $t_W = 1.03\,\mathrm{nm}$ (graphene) or $\sim 9.35\,\mathrm{GW\,m^{-2}}$ for $t_W = 1.10\,\mathrm{nm}$ (GO).

**Interfacial thermal coupling at the bio–nano interface**. To quantify the role of intercalated water layers in enhancing the thermal dissipation and model the process of thermal energy transport across the graphene–lipid bilayer interface, we need to determine the key parameter for the interfacial thermal coupling, that is, the ITC or namely the Kapitza conductance, since the convection motion is negligible in this system. Here we consider the graphene–water–lipid hybrid as an integrated system in discussing the thermal coupling between graphene and lipids, where the interface is manifested by the presence of intercalated water layers. Our thermal relaxation simulation results show that the value of ITC increases with $t_W$ (Fig. 4a). To account for more realistic biological systems, we carried out additional MD simulations for a lipid bilayer membrane with potassium channel proteins (KcsA; Supplementary Fig. 3). The results show that the ITC of graphene–lipid contact ranges from 13.7 to $49.10\,\mathrm{MW\,m^{-2}\,K^{-1}}$, while that for graphene–lipid interface with proteins ranges from 15.8 to $41.1\,\mathrm{MW\,m^{-2}\,K^{-1}}$. This result suggests that the presence of proteins does not significantly

change the value of ITC. Previous studies reported that hydrogen bonds between proteins and intercalated water molecules are responsible for a promotion of interfacial thermal transport, which explains the fact that the protein–water interface (with hydrogen bonds) has a higher ITC of $100–300\,\mathrm{MW\,m^{-2}\,K^{-1}}$ than that for the octane–water interface (without hydrogen bonds), that is, $65\,\mathrm{MW\,m^{-2}\,K^{-1}}$ (refs 24–28). In our model, however, this effect is negligible because of the limited contact between intercalated water layer and protein.

The thermal conductance of lipid–graphene interface calculated here is on the order of $10\,\mathrm{MW\,m^{-2}\,K^{-1}}$, which is compared with the ITC values reported for other nanoscale solid–solid, solid–polymer, liquid-polymer and solid–liquid interfaces as summarized as a function of the interfacial energy $\Gamma$ in Fig. 4b. The 'soft' interface explored here between graphene/GO and lipid bilayer with intercalated water is in general at the lower bound of these interfaces with solids (Supplementary Note 1)[26–43]. This consistency arises because of the diffusive nature of interfacial thermal transport, and the similarity in the nature of interfacial intermolecular interactions (van der Waals forces, electrostatic interaction, hydrogen bonds and so on). Moreover, the quantitative difference in the ITCs measured for interfaces with graphene and GO suggests that by functionalizing the nanostructures, one could further tune the interfacial thermal coupling, by crosslinking the nanostructure with lipid bilayer for example[44].

**Predicting the thermal perturbation**. Excess heat perturbs the physiological state of cells and tissues. For example, tumour cells are susceptible to heat treatment, resulting in cell death above $43\,^{\circ}\mathrm{C}$ (ref. 45). Our results thus quantitatively predict the minimal power at which thermotherapic treatment could be efficiently achieved at the interface with graphene or GO. To obtain deeper insights into the heat transfer process at the bio–nano interface, we develop an analytical model for the thermal coupling of this bio–nano hybrid (Fig. 5a) based on the under-standings and results obtained from our MD simulations. The model consists of two resistors connected in serial for the graphene/water/lipid interface and the lipid bilayer, respectively. We assume a diffusive heat transport regime due to the weak inter-action across interfaces and strong scattering of heat carriers therein, which is similar as the interface between graphene and substrates we studied in earlier works[46,47]. Then the, interfacial heat transport can be described using the Fourier law, $J = \kappa A\,\mathrm{d}T/\mathrm{d}x$, where $J$ and $T$ are the heat flux and temperature, respectively. $\kappa$ is the thermal conductivity and $A$ is the area of

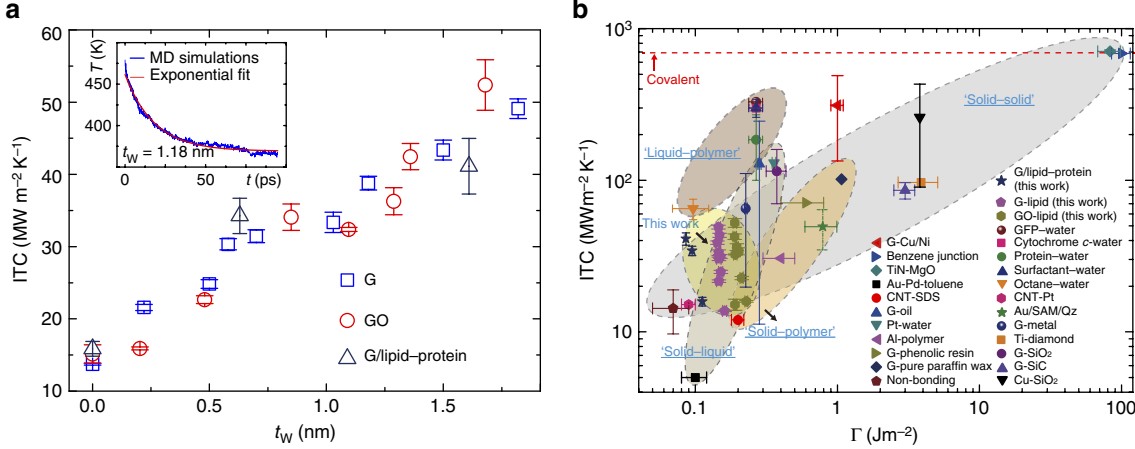

**Figure 4 | Interfacial thermal conductance (ITC) in typical systems.** (**a**) The ITC plotted as a function of $t_W$. The inset shows the exponential decay of temperature in the graphene layers after the heat pulse is removed, which is used to calculate the ITC in our thermal relaxation simulations. (**b**) A summary of ITC plotted as a function of the interfacial energy $\Gamma$ at solid–solid, solid–polymer, liquid–polymer and solid–liquid interfaces. The data are collected from the literature[26–43].

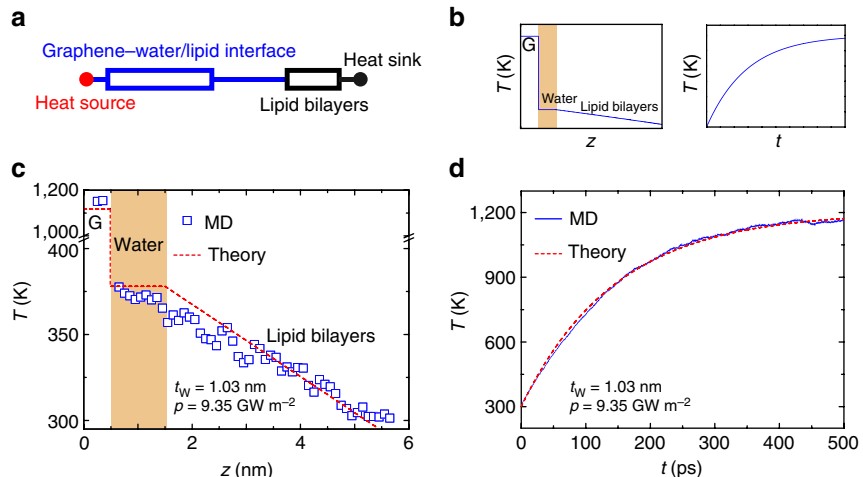

**Figure 5 | Model prediction of heat dissipation process across the bio–nano interface.** (**a**) A resistor-network model for the heat transfer pathway across the graphene–lipid membrane interface, which consists of two resistors representing the graphene–water/lipid interface and the lipid bilayer, respectively. (**b**) Steady-state temperature profile across the interface and time-dependent temperature evolution in the graphene layer calculated from the model, which agree well with our MD simulation results summarized in **c** and **d**.

interface. The steady-state temperature profile then be calculated from

$$T_G = T_W + J/AG_C \tag{1a}$$

$$T_W = T_e + J\delta/A\kappa_L \tag{1b}$$

$$T_L(z) = T_e + J(z_s - z)/A\kappa_L \tag{1c}$$

Here $G_C$ is the ITC we calculated from MD simulations for the graphene/water/lipid hybrid. $T_G$, $T_L$ and $T_W$ are the temperature of graphene/GO, lipid and intercalated water layer, respectively. We consider $T_W$ as a constant because MD simulation results show that temperature distribution in the intercalated water layer is uniform because of its high thermal conductivity ($\sim 0.61\,\mathrm{W\,m^{-1}\,K^{-1}}$) compared with the lipids ($\sim 0.12\,\mathrm{W\,m^{-1}\,K^{-1}}$), and the convectional contribution can be neglected. $z_s$ is the position of surface in the lipid bilayer. $\delta$ and $\kappa_L$ are the thickness and thermal conductivity of the lipid bilayer, respectively. $T_e = 300\,\mathrm{K}$ is the temperature of the water reservoir in contact with lipid bilayer on the other side, or the heat sink. By solving these set of Fourier's equations with boundary and

interface conditions, the temperature distribution across the whole hybrid, $T(x)$, is plotted in Fig. 5b, which shows an abrupt drop at the graphene–water interface because of the high interfacial thermal resistance, followed by a plateau in water and a temperature gradient in the lipid bilayer. With fitted values of $\kappa_L = 0.12\,\mathrm{W\,m^{-1}\,K^{-1}}$ and $G_C = 12.60\,\mathrm{MW\,m^{-2}\,K^{-1}}$, we can predict the temperature profile that agrees well with our MD simulation results (Fig. 5c). The results for the lipid bilayer with embedded KcsA protein are similar as the pure lipid bilayer case, indicating the same mechanism of interfacial thermal coupling (Supplementary Fig. 3).

While the graphene or GO layer is heated at a specific power density $p$, our diffusive model can also make predictions for the temperature evolution, which approaches a plateau as the heating process proceeds to the steady state (Fig. 5b). The evolution of temperature $T_g(t)$ in graphene at time $t$ follows the solution from the Fourier's equation using the lumped-parameter method (see Supplementary Note 2 for details)

$$T_g(t) = T_e + p/G_C - p/G_C\exp\left(-tG_C/c_g\rho_g d\right) \tag{2}$$

Here $d = 0.34$ nm is the nominal thickness of graphene defined by its van der Waals diameter, and $\rho_g$ and $c_g$ are the mass density and specific heat of graphene, respectively. This solution aligns perfectly with our MD simulation results with a reasonable set of parameters $\rho_g = 2.265$ g cm$^{-3}$, $c_g = 3N_A k_B = 2.1$ J g$^{-1}$ K$^{-1}$ and $G_C = 10.16$ MW m$^{-2}$ K$^{-1}$, as shown in Fig. 5d and Supplementary Fig. 5. These agreements further validate the predictability of our minimal model and the assumption therein. In practical applications, heat transport or leaking through bulk water outside of the confined region between graphene and the lipid membrane could also contribute to the heat dissipation (Supplementary Fig. 4). However, our analysis shows that the path across the bio–nano interface is the dominant one. Consequently, once the heating power and ITC are known for a specific biomedical device set-up, the targeted temperature rise in the nanodevice can be well predicted from our model with parameters extracted from MD simulations that can include the effects of interfacial chemistry and intermolecular interactions. On the other hand, the minimum power density for invasive thermal treatment of cell or tissues could also be predicted following this approach.

## Discussion

In this work, we analysed the molecular structures and diffusivity of interfaces between graphene (GO) and lipid membrane by performing atomistic simulations. Structured water was identified near the graphene or GO layers with a maximum thickness of 1 nm, which reduces the molecular diffusivity of water and enhances thermal dissipation from the nanodevice. By considering the conditions for bulk-like molecular diffusivity and efficient electrical signal communication, energy exchanges between graphene and lipid bilayer, this dimension defines a critical distance in the design of biomedical applications. The thermal conductance of this bio–nano interface is on the order of 10 MW m$^{-2}$ K$^{-1}$ that is within the typical range of conductance of soft nanoscale interfaces. Thermal coupling processes at this interface can be captured by modelling the graphene/water/lipid interface and lipid bilayer as a network of thermal resistors. This analytical model yields results that agree well with our MD simulations for the temperature profile and temperature rise in the nanostructure, which can be used to evaluate the critical value of power generation in nanodevices before the accumulated heat disturbs living tissues. These findings and models lay the ground for future rational designs of wearable and implantable devices for biosensing and other biomedical applications such as thermotherapic treatments.

## Methods

**Atomic structures.** We construct the graphene–lipid bilayer interface by placing the bilayer at a certain distance from graphene. Water molecules are added in between, which could enter the spaces in 1-palmitoyl-2-oleoyl-sn-glycero-3-phos-phocholine (POPC) lipid bilayer with a depth of ∼1 nm. The other side of the lipid bilayer is also immersed in water, modelled through a thin layer of water. For GO, we construct hydroxyl-functionalized graphene on both sides of the sheet with concentrations $c = n_O/n_C = 20\%$ following recent experimental evidences[48]. Here $n_O$ and $n_C$ are the numbers of oxygen-rich groups and carbon atoms. The spatial distributions of hydroxyl groups are sampled randomly in the oxidized region. This model has been successfully used to predict the wetting behaviours of graphene–water and GO–water interfaces[20,49,50]. A two-dimensional supercell is used with periodic boundary conditions along the interface, with lateral dimensions of 5.36 and 5.14 nm, and a open boundary condition is used in the direction across the bilayer.

**MD simulations.** All classical MD simulations are performed using the large-scale atomic/molecular massively parallel simulator package[51]. The thermodynamics, structural and mechanical properties of lipid bilayers and proteins in the cell membrane have been successfully explored using this approach[52–55]. However, limitations of the current model include the disregard of salt and pH effects, which is difficult because of the low concentration of ions and limited system size in fully atomistic simulations. The interatomic interactions for graphene and GO are described using the all-atom optimized potential for liquid simulations, which can capture essential many-body terms in interatomic interactions, including bond stretching, bond angle bending, dihedial angle bending and van der Waals[41]. The CHARMM36 force field is used for the lipid bilayer and the TIP3P model is used for water[56]. The SHAKE algorithm is applied for the stretching terms between oxygen and hydrogen atoms to avoid numerical integration of hydrogen-related high-frequency vibrations that require a much shorter time step. The interaction between water, graphene and the lipid bilayer includes both van der Waals and electrostatic terms. The former one is described by the 12–6 Lennard–Jones potential $4\varepsilon[(\sigma/r)^{12} - (\sigma/r)^6]$ at an interatomic distance $r$. The van der Waals forces are truncated at 1 nm and the long-range Coulomb interactions are computed using the particle–particle particle–mesh algorithm[57]. The time step is set to 0.1 fs to assure energy conservation in the absence of thermostat coupling. The whole system is equilibrated at 300 K and 1 atm before the heat transfer simulations are carried out.

**Calculation of molecular diffusion coefficient.** The molecular diffusion coefficient $D$ is calculated from the molecular trajectories of water using Einstein's definition that relates mean-square distance to $D$ as $D = \lim_{t \to \infty} < |r(t) - r(0)|^2 > /2d_i t$. Here $d_i$ is the dimension of space for diffusion, $t$ is the diffusing time and $< \dots >$ is the ensemble average that is implemented by averaging $D$ obtained from multiple independent simulation runs.

**Simulation of heat dissipation.** We simulate the heat dissipation process by scaling atomic velocities in graphene, while keeping the water reservoir at the other side of the lipid bilayer at 300 K by coupling to a Berendsen thermostat. Heat dissipation at the interface undergoes two processes. Temperature in graphene, water and lipid bilayers increases gradually with time in the transient process before the system reaches the steady state with a constant temperature profile. The temperature distribution across the whole system is calculated by averaging within thin slabs of 0.1 nm thick and time intevals of 1 ps. In thermal relaxation simulations, a 100 ps pulse heat is injected to graphene to raise the temperature to ∼500 K. We then extract the thermal relaxation time $\tau$ at the interface and calculate the ITC or Kapitza conductance $G_K$ via $G_K = C/\tau A$, where $C$ is the heat capacity of graphene and $A$ is the area of interface[46].

**Data availability.** The data that support the findings of this study are available from the corresponding authors upon request.

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

## Acknowledgements

This work was supported by the MIT-China seed fund, the National Natural Science Foundation of China (Grant No. 11472150), and the National Key Basic Research Program of China (Grant No. 2015CB351900). This work was also supported by DOD-ONR (Grant No. N00014-16-1-233), ONR PECASE (Grant No. N00014-10-1-0562), AFOSR FATE MURI (Grant No. FA9550-15-1-0514), DARPA, the MIT Energy Initiative, and in part by the MRSEC Program of the National Science Foundation under award number DMR-0819762. The simulations were performed on the Explorer 100 cluster system of Tsinghua National Laboratory for Information Science and Technology.

## Author contributions

Z.Q., M.J.B. and Z.X. designed the research. Y.W. and Z.X. implemented the model and analysis tools, carried out the simulations, and collected the data. Y.W., Z.Q. and Z.X analysed and interpreted the data. All the authors contributed to writing the paper.

## Additional information

**Competing financial interests:** The authors declare no competing financial interests.

