## [Peer review file · Nature Communications]

Reviewers' Comments:

Reviewer #1 (Remarks to the Author)

The manuscript reports the results of large-scale all-atom MD simulations to investigate thermal conduction at the interface between a lipid bilayer and graphene or graphene oxide (GO), including a layer of water between them that ranges in size from 0 to 2 nm, since experiments indicate that such a layer could be on the nm scale. The lipid bilayer is meant to model a cell membrane, as the authors point out in the manuscript and in figure 1. Graphene (or GO) is heated to a constant power density and thermal transport is monitored in the simulation. The authors find that above a "critical" power density the temperature in the membrane begins to rise. The value of this "critical" power density depends on the thickness of the water layer. The results appear to establish roughly 1 GW m^{-2} as the power density below which there is little chance of thermal buildup in the lipid bilayer (cell membrane). The results of the simulations that have been carried out may be of interest as there may potentially be applications to biological systems, but more needs to be done to make that case and substantial revision is required before the manuscript could be considered for publication in Nature Communications.

The model of a cell membrane used in the simulations is not realistic. No cell membrane is a pure lipid bilayer as illustrated in figure 1(a). A cell membrane may well be something like 50% by volume protein (channels, receptors, etc.). Therefore the graphene will often interface about as much with proteins as lipid bilayer for a realistic cell membrane. Again, water would likely lie at the interface of the graphene and protein. To address applications to cell membranes the authors should provide estimates of how the presence of protein influences their conclusions of critical power density in their discussion. While the graphene-protein interfacial boundary conductance may not be known, the water-protein boundary conductance is known (see below) and thermal transport coefficients for protein molecules are known. The values for the thermal transport coefficients that have been calculated ("Vibrational energy transfer and heat conduction in a protein," J. Phys. Chem. B 107, 1698 - 1707 (2003)) and measured ("Protein Thermal Conductivity Measured in the Solid State Reveals Anharmonic Interactions of Vibrations in a Fractal Structure," J. Phys. Chem. Lett. 5, 1077 - 1082 (2014)) for proteins are in good agreement over a wide range of temperature. For the manuscript to have any impact concerning application to cells, the authors need to address proteins as well as lipids and include their role in, for example, the temperature profile at a given power density, discussed on page 7, and the critical power density, discussed on pages 7 and 8.

On page 6 (and summarized again on page 8) the authors point out that their calculated value of the lipid-graphene interfacial thermal conductance is on the order of $10 \text{ MW m}^{-2} \text{ K}^{-1}$. (They use the value $12.70 \text{ MW m}^{-2} \text{ K}^{-1}$ to predict a temperature profile with a simple model.) They claim that this value is in the typical range of "soft" nanoscale interfaces. In some cases they are right, but in other cases, in particular for protein-water interfaces, the values are much larger. The authors should compare with previously calculated results for protein-water interfacial thermal conductance, for example "Vibrational energy flow through the green fluorescent protein-water interface: Communication maps and thermal boundary conductance," J. Phys. Chem. B 18, 7818 - 7826; "Heat transfer in protein-water interfaces," Phys. Chem. Chem. Phys. 12, 1670 - 1627 (2010); "Vibrational energy flow across heme-cytochrome c and cytochrome c-water interfaces," Theor. Chem. Acc. 133, 1504 (2014). These three studies focus on the interfacial thermal conductance of several proteins and water, with values ranging from 100 to 300 $\text{MW m}^{-2} \text{ K}^{-1}$ near 300 K, which should be added to figure 4b for comparison. The authors of those studies compared the results with the interfacial thermal conductance of other "soft" materials, which can also be quite large and are also worth mentioning. Such large values of interfacial thermal conductance are found when hydrogen bonding between soft materials is involved, and hydrogen bonds are of course ubiquitous in biological systems.

The authors obtain their values for the interfacial thermal conductance using classical molecular dynamics simulations. It would be useful if they would point out any possible limitations of this method. Could quantum mechanical effects be important?

Reviewer #2 (Remarks to the Author)

The authors discuss a temperature elevation in a biological membrane which is in contact with a heated graphene, where both are separated by a thin layer of water. Using MD simulations, the authors show under which conditions the temperature in the bilayer reaches certain values. The temperature rise increases in a nonlinear way with the power generated in graphene and decreases linearly with the water layer thickness. A simple serial network model is used to explain the observations.

In principle, the work is of interest, but a deeper understanding of the observations prevents the reader to build a quality picture of the studied phenomena. For example, why is the dependence on power nonlinear? Is it because of a different regime of heat transfer through water? Authors talk about dissipation. Do they mean dissipation along the heat gradient? To explain the above nonlinearity, one could naively think that water starts to perform some convective motion. However, this would have to be tested by simulations. Overall, better understanding of these observations is necessary before the paper could be considered for Nature Communications.

I also have some technical comments:

- 1) Related study was recently performed in: A. Titov, P. Kral and R. Pearson, Sandwiched Graphene-Membrane Superstructures, ACS Nano 4, 229 (2010). The authors should cite this work, where graphene is differently positioned within the membrane.
- 2) Overall, the English is weak at certain places and could be strengthened.

Reviewers' comments:

Reviewer #1 (Remarks to the Author):

The manuscript reports the results of large-scale all-atom MD simulations to investigate thermal conduction at the interface between a lipid bilayer and graphene or graphene oxide (GO), including a layer of water between them that ranges in size from 0 to 2 nm, since experiments indicate that such a layer could be on the nm scale. The lipid bilayer is meant to model a cell membrane, as the authors point out in the manuscript and in figure 1. Graphene (or GO) is heated to a constant power density and thermal transport is monitored in the simulation. The authors find that above a "critical" power density the temperature in the membrane begins to rise. The value of this "critical" power density depends on the thickness of the water layer. The results appear to establish roughly 1 GW m^{-2} as the power density below which there is little chance of thermal buildup in the lipid bilayer (cell membrane). The results of the simulations that have been carried out may be of interest as there may potentially be applications to biological systems, but more needs to be done to make that case and substantial revision is required before the manuscript could be considered for publication in Nature Communications.

Reply: Thank you for the kind review and the detailed comments. We made substantial changes to the manuscript based on your suggestions. Please see below the details of our revisions. The edits we have made to our revised manuscript has been highlighted in blue.

*The model of a cell membrane used in the simulations is not realistic. No cell membrane is a pure lipid bilayer as illustrated in figure 1(a). A cell membrane may well be something like 50% by volume protein (channels, receptors, etc.). Therefore the graphene will often interface about as much with proteins as lipid bilayer for a realistic cell membrane. Again, water would likely lie at the interface of the graphene and protein. To address applications to cell membranes the authors should provide estimates of how the presence of protein influences their conclusions of critical power density in their discussion. While the graphene-protein interfacial boundary conductance may not be known, the water-protein boundary conductance is known (see below) and thermal transport coefficients for protein molecules are known. The values for the thermal transport coefficients that have been calculated ("Vibrational energy transfer and heat conduction in a protein," *J. Phys. Chem. B* 107, 1698 - 1707 (2003)) and measured ("Protein Thermal Conductivity Measured in the Solid State Reveals Anharmonic Interactions of Vibrations in a Fractal Structure," *J. Phys. Chem. Lett.* 5, 1077 - 1082 (2014)) for proteins are in good agreement over a wide range of temperature. For the manuscript to have any impact concerning application to cells, the authors need to address proteins as well as lipids and include their role in, for example, the temperature profile at a given power density, discussed on page 7, and the critical power density, discussed on pages 7 and 8.*

Reply: Thank you for this valuable suggestion, which we really appreciated. Lipid bilayers were chosen in our study as it represents a simple and representative material to interface with graphene through the intercalated water layers at the interface with biological tissues. As reported in the references mentioned above (which have been added to the citation list), the presence of protein may modify the heat transport and dissipation processes through the interface. To address this concern, we carried out a set of MD simulations for potassium channel proteins (KcsA) embedded in the lipid bilayer. Our results show that the nature of interfacial coupling remains the same, but quantitatively the interfacial thermal conductance and dissipation efficiency are altered. We revised the manuscript by adding these new results, including the temperature profiles and critical power densities (these are included in the **Supplementary Materials**).

Changes made:

(1) On pages 6 and 8, discussion on the effect of proteins at the interface with lipid bilayers was added.

“To account for more realistic biological systems, we carried out additional MD simulations for a lipid bilayer membrane with potassium channel proteins (KcsA) (**Fig. S3**). The results show that the ITC of graphene-lipid contact ranges from 13.7 to $49.10 \text{ MWm}^{-2}\text{K}^{-1}$, while that for graphene-lipid interface with proteins ranges from 15.8 to $41.1 \text{ MWm}^{-2}\text{K}^{-1}$. This result suggests that the presence of proteins does not significantly change the value of ITC. Previous studies reported that hydrogen bonds (HBs) between proteins and intercalated water molecules are responsible for a promotion of interfacial thermal transport, which explains the fact that the

protein-water interface (with HBs) has a higher ITC of 100-300 $\text{MWm}^{-2}\text{K}^{-1}$ than that for the octane-water interface (without HBs), *i.e.* 65 $\text{MWm}^{-2}\text{K}^{-1}$.^{24,25,26,27,28} In our model, however, this effect is negligible because of the limited contact between intercalated water and protein.”

“The results for the lipid bilayer with embedded protein are similar, indicating the same mechanism of interfacial thermal coupling (**Fig. S3**).”

(2) References 24 and 25 were added.

24. Yu X, Leitner DM. Vibrational energy transfer and heat conduction in a protein. *J. Phys. Chem. B* **107**, 1698-1707 (2003).

25. Foley BM, *et al.* Protein thermal conductivity measured in the solid state reveals anharmonic interactions of vibrations in a fractal structure. *J. Phys. Chem. Lett.* **5**, 1077-1082 (2014).”

(3) **Figure S3** was added to the **Supplementary Materials**.

Figure S3. (a) Molecular model of a graphene/cell membrane interface by considering a potassium channel protein embedded in the lipid bilayer, which contains the protein, lipid molecules, water molecules, graphene and 13 chloride ions to ensure electroneutrality of the system.¹ In the MD simulations, a two-dimensional supercell is used with periodic boundary conditions along the interface, with lateral dimensions of 7.20 and 7.25 nm, and an open boundary condition is used in the direction across the bilayer. The CHARMM36 forcefield is used for the protein, and the other setup is the same as that for graphene/lipid bilayer shown in “Method” parts.¹ (b) Temperature rise in the lipid bilayer after a heat flux with power density ρ is injected into graphene. (c-d) The evolution of temperature in the MD simulations, which reaches a steady state after ~ 2 ns. This result shows that the protein has a slightly higher temperature than the lipid membrane. This inhomogeneity may explain the lower critical power predicted for the graphene/lipid-protein interface (panel b) than the graphene/lipid interface but the difference is moderate (**Fig. 3a**).

On page 6 (and summarized again on page 8) the authors point out that their calculated value of the lipid-graphene interfacial thermal conductance is on the order of $10 \text{ MW m}^{-2} \text{ K}^{-1}$. (They use the value $12.70 \text{ MW m}^{-2} \text{ K}^{-1}$ to predict a temperature profile with a simple model.) They claim that this value is in the typical range of "soft" nanoscale interfaces. In some cases they are right, but in other cases, in particular for protein-water interfaces, the values are much larger. The authors should compare with previously calculated results for protein-water interfacial thermal conductance, for example "Vibrational energy flow through the green fluorescent protein-water interface: Communication maps and thermal boundary conductance," *J. Phys. Chem. B* **18**, 7818 - 7826; "Heat transfer in protein-water interfaces," *Phys. Chem. Chem. Phys.* **12**, 1670 - 1627 (2010); "Vibrational energy flow across heme-cytochrome *c* and cytochrome *c*-water interfaces," *Theor. Chem. Acc.* **133**, 1504 (2014). These three studies focus on the interfacial thermal conductance of several proteins and water, with values ranging from 100 to $300 \text{ MW m}^{-2} \text{ K}^{-1}$ near 300 K, which should be added to figure 4b for comparison. The authors of those studies compared the results with the interfacial thermal conductance of other "soft" materials, which can also be quite large and are also worth mentioning. Such large values of interfacial thermal conductance are found when hydrogen bonding between soft materials is involved, and hydrogen bonds are of course ubiquitous in biological systems.

Reply: Thank you for bringing these important references to our attention. We added these data for protein-water interfaces into our discussion. The effect of hydrogen bonds is important for the protein-water contact, which explains the reported high interfacial thermal conductance. However, this effect is negligible in this work because of the limited contact between intercalated water and proteins. We added relevant discussion to the revised manuscript.

Changes made:

(1) On page 6, See the changes in our reply for your previous comment.

(2) References 26-28 were added.

“26. Agbo JK, Xu Y, Zhang P, Straub JE, Leitner DM. Vibrational energy flow across heme-cytochrome *c* and cytochrome *c*-water interfaces. *Theor. Chem. Acc.* **133**, 1-10 (2014).

27. Xu Y, Leitner DM. Vibrational energy flow through the green fluorescent protein-water interface: Communication maps and thermal boundary conductance. *J. Phys. Chem. B* **118**, 7818-7826 (2014).

28. Lervik A, Bresme F, Kjelstrup S, Bedeaux D, Miguel Rubi J. Heat transfer in protein-water interfaces. *Phys. Chem. Chem. Phys.* **12**, 1610-1617 (2010).”

(3) **Figure 4b** was modified by adding these data as ‘liquid-polymer’ interfaces.

The authors obtain their values for the interfacial thermal conductance using classical molecular dynamics simulations. It would be useful if they would point out any possible limitations of this method. Could quantum mechanical effects be important?

Reply: Although this is the first study of interfacial thermal coupling between graphene (oxides) and cell membranes, there have been many previous studies on graphene-metal/dielectric and polymer-substrate interfaces (Refs. 52-55). Classical molecular dynamics simulations have been validated for those systems through comparison with experimental data. The thermodynamics of biological systems with proteins, lipid bilayers and water has also been explored through classical molecular dynamics. To assess more directly the potential quantum effects in this system, we use path-integral molecular dynamics (PIMD) to calculate the vibrational spectrum of graphene/copper interface (see **Figure R1** below) because the full-atom PIMD simulations for the graphene/water/lipid bilayer system is too expensive. We find that the quantum effect on thermodynamics becomes negligible at room temperature, which is the condition we focus on in the current study. Considering the much higher Debye temperature of graphene and copper than lipids and proteins, we conclude that at the condition we consider in this study, classical MD simulations are reliable. However,

possible limitations of the current model include the disregard of salt and pH effects, which is difficult because of the low concentration of ions and limited system size in simulations.

Figure R1 Comparison between calculated vibrational spectra of graphene/copper hybrid system using classical and quantum-mechanical dynamics simulations at temperature $T = 20, 100$ and 300 K. The consistence at temperature higher than 100 K shows that the quantum nuclei effect is insignificant at room temperature, as we consider in this work. The quantum nuclei effect is modeled through the path-integral molecular dynamics (PIMD) simulations in a ring-polymer representation.

Changes made:

(1) On page 10, in the Methods section, we added discussion on the validity of our classical MD approach.

“The thermodynamics, structural and mechanical properties of lipid bilayers and proteins in the cell membrane have been successfully explored using this approach.^{52, 53, 54, 55} However, limitations of the current model include the disregard of salt and pH effects, which is difficult because of the low concentration of ions and limited system size in full-atom simulations”

(2) References 52-55 were added.

“52. Harding JH, Duffy DM, Sushko ML, Rodger PM, Quigley D, Elliott JA. Computational techniques at the organic-inorganic interface in biomineralization. *Chem. Rev.* **108**, 4823-4854 (2008).

53. Baron R, Trzesniak D, de Vries AH, Elsener A, Marrink SJ, van Gunsteren WF. Comparison of thermodynamic properties of coarse-grained and atomic-level simulation models. *ChemPhysChem* **8**, 452-461 (2007).

54. Lindahl E, Edholm O. Mesoscopic undulations and thickness fluctuations in lipid bilayers from molecular dynamics simulations. *Biophys. J.* **79**, 426-433 (2000).

55. Kraszewski S, Tarek M, Treptow W, Ramseyer C. Affinity of C₆₀ neat fullerenes with membrane proteins: A computational study on potassium channels. *ACS Nano* **4**, 4158-4164 (2010).”

Reviewer #2 (Remarks to the Author):

The authors discuss a temperature elevation in a biological membrane which is in contact with a heated graphene, where both are separated by a thin layer of water. Using MD simulations, the authors show under which conditions the temperature in the bilayer reaches certain values. The temperature rise increases in a nonlinear way with the power generated in graphene and decreases linearly with the water layer thickness. A simple serial network model is used to explain the observations.

In principle, the work is of interest, but a deeper understanding of the observations prevents the reader to build a quality picture of the studied phenomena. For example, why is the dependence on power nonlinear? Is it because of a different regime of heat transfer through water? Authors talk about dissipation. Do they mean dissipation along the heat gradient? To explain the above nonlinearity, one could naively think that water starts to perform some convective motion. However, this would have to be tested by simulations. Overall, better understanding of these observations is necessary before the paper could be considered for Nature Communications.

Reply: Thank you for these valuable comments. Based on our model and MD simulations results, we added more discussion on the data in this revision.

(1) Nonlinear dependence temperature rise on power density, contribution of convection motion, and the regime of heat transfer.

The temperature gradient in the intercalated water is very small (a few Kelvin) so it is difficult to induce convection motion of water. According to the trajectories of water molecules during the non-equilibrium MD simulations for heat dissipation (**Figure R2** below), we conclude that the water molecules are diffusing instead of following convection (thermal gradient is along the Z direction across the graphene-lipid interface). Consequently, heat transfer is in the conduction regime. However, the structure and diffusivity of water changes with the temperature, or the power density applied. The modest deviation from linearity may arise from these changing properties of intercalated water, which is more significant for GO because of its stronger interaction with water (**Fig. 3a** and **S1a**),

Figure R2. Molecular trajectories of intercalated water at different thickness t_w , under a thermal gradient applied across the interface, along the Z direction. Clearly the convection motion is negligible and the water molecules are diffusing in the intercalated water layer.

(2) Definition of heat dissipation.

Yes, the heat dissipation means heat energy leaking from graphene/GO sheets to the sink through their interface with intercalated water/lipid bilayers, along the direction of temperature gradient.

(3) In addition to exclude the contribution from convection from the heat conduction process, another important understanding from the analytical model is that the assumption of a diffusive heat transport mechanism can quantitatively reproduce the MD simulation results, where parameters such as the resistance can be determined accordingly. The physics behind the validity lies on the weak interaction of lipid/graphene interfaces and strong scattering of the heat carriers therein.

Changes made:

On pages 5, 6 and 8, all the abovementioned issues were addressed in this revision.

“As shown in **Fig. 3a**, at a low heating power density, e.g. $p \sim 1 \text{ GWm}^{-2}$, the heat generated can be efficiently dissipated across the interface, along the thermal gradient.”

“The modest deviation from linearity may arise from the change in the structure and diffusivity of intercalated water at different heating power density, which is more significant for GO because of its stronger interaction with water (**Fig. 3a** and **S1a**).”

“To quantify the role of intercalated water layers in enhancing the thermal dissipation and model the process of thermal energy transport across the graphene-lipid bilayer interface, we need to determine the key parameter for the interfacial thermal coupling, that is, the ITC or namely the Kapitza conductance since the convection motion is negligible in this system.”

“We consider T_w as a constant because MD simulation results show that temperature distribution in the intercalated water layer is uniform because of its high thermal conductivity ($\sim 0.61 \text{ Wm}^{-1}\text{K}^{-1}$) compared to the lipids ($\sim 0.12 \text{ Wm}^{-1}\text{K}^{-1}$), and the convective contribution can be neglected”

I also have some technical comments:

1) Related study was recently performed in: A. Titov, P. Kral and R. Pearson, Sandwiched Graphene-Membrane Superstructures, ACS Nano 4, 229 (2010). The authors should cite this work, where graphene is differently positioned within the membrane.

Reply: Thank you for introducing this interesting and relevant study. We added discussion on this work in the revision.

Changes made:

(1) On page 2, in the Introduction section, we discussed this reference.

“Although recent studies show that micro- and nanometer-sized graphene sheets could enter the hydrophobic interior of biological membranes,^{16, 17} our work here focuses on the application of graphene-membrane contacts for sensing and actuation applications, where the graphene sheet covers the cell surface.”

(2) Refs. 16 and 17 were added.

“16. Titov AV, Král P, Pearson R. Sandwiched graphene-membrane superstructures. *ACS Nano* 4, 229-234 (2010).

17. Li Y, *et al.* Graphene microsheets enter cells through spontaneous membrane penetration at edge asperities and corner sites. *Proc. Natl. Acad. Sci. USA* 110, 12295-12300 (2013)”

2) *Overall, the English is weak at certain places and could be strengthened.*

Reply: Thank you for this suggestion. We polished our manuscript thoroughly.

Reviewers' Comments:

Reviewer #1 (Remarks to the Author)

The authors have substantially revised and improved the manuscript. They have addressed all of my concerns with the original version. The additional study they present with the ion channels in particular makes a more convincing case and will broaden the impact of the work that is reported. Publication in Nature Communications is now recommended.

Reviewer #2 (Remarks to the Author)

The authors have responded in a satisfactory way to the questions raised. I recommend the paper for publication.